# Mechanical Thrombectomy for Acute Ischemic Stroke in Patients with Malignancy: A Systematic Review

**DOI:** 10.3390/jcm11164696

**Published:** 2022-08-11

**Authors:** Athina-Maria Aloizou, Daniel Richter, Jeyanthan Charles James, Carsten Lukas, Ralf Gold, Christos Krogias

**Affiliations:** 1Department of Neurology, St. Josef-Hospital, Ruhr University Bochum, 44892 Bochum, Germany; 2Institute of Neuroradiology, St. Josef-Hospital Bochum, Ruhr University Bochum, 44892 Bochum, Germany

**Keywords:** ischemic stroke, cancer-related stroke, thrombectomy, malignancy

## Abstract

Background: Ischemic stroke is a common occurrence in patients with concomitant malignancy. Systemic thrombolysis is often contraindicated in these patients, and mechanical thrombectomy (MT) is the preferred method of intervention. This review aims to collect the available data on the safety and efficacy of MT in cancer patients (CPs).Methods: The PubMed/MEDLINE and SCOPUS databases were systematically searched for studies assessing safety (mortality, intracranial hemorrhage) and efficacy (reperfusion, functional outcome) indices in CPs receiving MT. Potentially relevant parameters examined in solitary studies were also extracted (e.g., stroke recurrence, brain malignancy).Results: A total of 18 retrospective studies of various methodologies and objectives were identified. Rates of in-hospital mortality, intracranial hemorrhage of any kind, reperfusion rates, and discharge condition did not seem to present any considerable differences between CPs and patients without cancer. On the contrary, 90-day mortality was higher and 90-day functional independence was lower in CPs. Three studies on cancer-related stroke (no other identifiable etiology and high D-dimer levels in the presence of active cancer) showed constant tendencies towards unfavorable conditions. Conclusions: Per the available evidence, MT appears to be a safe treatment option for CPs. It is still unclear whether the 90-day mortality and outcome rates are more heavily influenced by the malignancy and not the intervention itself, so MT can be considered in CPs with prospects of a good functional recovery, undertaking an individualized approach.

## 1. Introduction

An acute ischemic stroke (AIS) is a neurological emergency stemming from the acute obstruction of blood flow in a cerebral territory. A plethora of predisposing health conditions are known, for instance, atrial flatter, carotid disease, diabetes, and smoking, but the presence of an underlying malignancy also seems to increase the risk of AIS, with 1 in 20 hospitalized AIS patientshaving a concurrent malignancy [1]. AIS can occur in cancer patients due to several different factors, such as hypercoagulability and platelet disorders, or as a result of cancer therapies, namely chemotherapy [2].

The recanalization treatment of AIS consists of intravenous thrombolysis (IVT) and mechanical thrombectomy (MT) [3]. Though cancer per se is no contraindication for administering IVT, contraindications, such as low platelet counts, recent surgeries, and active hemorrhages, are more often encountered in cancer patients (CPs), so MT often remains the only recanalization modality when applicable [4]. As an interesting temporal trends study showed, though both methods demonstrated an increase in time, cancer patients received MT at similar rates compared to non-cancer patients, whereas IVT was administered in about two-thirds of the cases of cancer patients compared to non-cancer ones [1]. There is a general paucity of high-quality data on whether these treatment modalities are safe and effective in patients with cancer, and most guidelines do not make any particular mention of malignancy. Only recently did the guidelines of the American Heart Association include particular clauses for cancer patients, recommending IVT for cancer patients with a reasonable life expectancy of more than 6 months, excluding those with structural gastrointestinal malignancies or brain malignancies [5]. However, no such recommendations exist for MT, since only scarce data were available until recently, with more studies emerging only in the last four years, and with cancer patients being generally excluded from the big clinical studies that established the efficacy of MT [6].

This systematic review aimed to summarize the studies that evaluated the safety and efficacy of MT in IS patients with active cancer, past cancer, or cancer-related ischemic strokes, in an attempt to facilitate decision-making for clinicians handling such patients in an emergency setting, and raise awareness for complications after the intervention that may require a heightened level of caution with these patients.

## 2. Search Algorithm

The PubMed/MEDLINE and SCOPUS databases were searched for studies assessing the efficacy and/or safety of MT in IS patients with cancer, active or past, with or without comparison to other patient subgroups. The words “Neoplasms”, “Thrombectomy”, and “Stroke” were used as MeSH terms, with “malignancy/ies”, “ischemic stroke”, “cerebral ischemia”, “mechanical thrombectomy”, and “endovascular treatment” added as free-text words for the search algorithm with the respective AND/OR Boolean operators. The search, performed on the 4th of April 2022 yielded 160 results from PubMed and 209 results from SCOPUS, from 1989 to 2022. They were then assessed firstly through title and then through abstract, and those deemed potentially fitting were assessed in their full text. Case reports, case series, and studies not mentioning the respective metrics for MT, as well as any reviews/meta-analyses, were excluded. Studies primarily assessing IVT but providing relevant metrics for the subgroup of IVT+MT were also included. The references of the retrieved articles and reviews/meta-analyses were scanned for any studies omitted by the original search algorithm. A total of 18studies were finally included. The review adhered to the PRISMA recommendations for systematic reviews and the step-by-step process (PRISMA Flowchart) can be seen in Figure 1.

The studies were assessed for rates of complications, such as intracranial bleeding, and mortality, both peri-interventionally and longitudinally, and outcomes (reperfusion rates, functional outcomes). Parameters controlled in individual studies that were also deemed relevant and of clinical importance were noted as well.

## 3. Results

### 3.1. Study Types, Included Patients, Baseline Characteristics

All of the included studies were retrospective analyses of either files/databases of patients that received acute treatment for AIS [6,7,8,9,10,11,12,13,14,15,16,17,18,19], or analyses of hospital records with the use of diagnostic codes, e.g., ICD codes for AIS and malignancy [20,21,22]. One study was a retrospective analysis of cancer patients from the MR CLEAN thrombectomy study registry [23].

There was a certain amount of discrepancy regarding the patients included in the studies. Most studies included patients with AIS and an active malignancy (AM), defined as patients with a diagnosis of current or metastatic cancer, those undergoing treatment for cancer, or who refused the treatment [6,7,10,11,13,14,15,16,18,24], and at times patients with a diagnosis made during or after the AIS-hospitalization [10,14,15]. This definition was occasionally restricted to a diagnosis within a year or 6months, or cancer treatment within 1–6 months of the ischemic event [7,11,18,23]. Others utilized ICD codes pertaining to active and metastatic malignancies [20,21,22], while certain studies analyzed all patients with any history of cancer [9], or did not provide a solid definition of active/underlying cancer [17]. Entities such as myelodysplastic syndromes and non-invasive basal cell carcinoma were, in certain studies, specifically mentioned as not considered within the definition of active cancer [23], while others excluded patients with hematological or intracranial malignancies/metastases [14,15,21]. Intracranial tumors are usually considered a contraindication for recanalization procedures [5], and, as such, these patients might have been excluded in the majority of studies without specific mention.

Three studies assessed patients with “cancer-related stroke” (CRS), not just AIS patients with a concomitant malignancy. This entity has been recently introduced in stroke discussions and pertains to AIS that directly stems from the underlying malignancy, usually in the presence of hypercoagulability. In these studies, CRS was defined as AIS of unknown/cryptogenic etiology after exclusion of cardioembolism and large artery atherosclerosis, with elevated D-dimer levels and/or diffusion-restricted lesions in multiple vascular territories, in patients with active cancer [12,14,19]. Taking into consideration the distinct nature of this type of AIS, the results of the studies assessing CRS will be presented in a different section.

The included studies, along with study characteristics and numbers of included cancer patients MT, are shown in Table 1.

Regarding baseline characteristics (Table 2), patients allocated in the malignancy groups were often found to be younger [13,18] and with lower rates of common stroke risk factors, such as atrial fibrillation, hyperlipidemia, and diabetes [7,10,11,12,13,14,16,18,20,21,23,25]. No study reported higher comorbidity rates in CPs; only one reported higher smoking rates in CPs [23], though smoking is also considered a major malignancy risk factor. Cancer patients were also more often found to have increased pre-stroke disability (modified Rankin Scale-mRS score ≥2) [23]. A few studies compared baseline values of biochemical markers commonly associated with cancer and cancer-related stroke [2], and found higher mean CRP [13,14] and D-dimer values [18], higher leukocyte counts [14], lower hemoglobin [14,18], and lower platelet counts [14,15]. Higher D-dimer values also correlated with unfavorable outcomes [14]. Respiratory, genitourinary, pancreatic, colorectal, and breast cancers were the most common cancer types reported [6,17,18,21,23], with adenocarcinomas also being mentioned as the most prevalent histological cancer type [7,8].

### 3.2. Mortality

#### 3.2.1. In-Hospital Mortality

In-hospital mortality (IHM) was assessed in 11 studies (Table 3, 3 CRS studies, see Section 3.6). Three studies found IHM significantly increased in active CPs compared to NCPs [11,22,23], and three studies found the difference between CPs and NCPs not statistically significant [10,17], with one of them reporting no difference in IHM between patients with active cancer, remote cancer, and no cancer history [15]. This was also shown in a study on IVT, which reported that in the subgroup of IVT+MT, IHM was not different between CP and NCP groups [20].

A different study reported that non-metastatic cancer did not affect IHM, regardless of treatment with MT or IVT, while metastatic cancer was not associated with IHM only in the patients receiving MT, with or without IVT; in this sense, MT led to a reduced IHM in these patients [21].

In a study of long-term prognosis, where no comparison was made between CPs and NCPs, 12 CPs were identified to have received MT; 5 of them (41.6%) deceased due to stroke-related reasons within 14 days [7].

#### 3.2.2. 90-Day and Long-Term Mortality

The 90-day mortality (90DM) was assessed in 14 studies (Table 3, 3 CRS studies, see Section 3.6) and was often found significantly increased in active CPs compared to NCPs [8,9,10,11,13,23]. In one study assessing long-term prognosis, 7 of 12 (58.3%) CPs deceased by the 90-day time-point, and 11 of 12 (91.6%) CPs treated with MT deceased due to stroke- or malignancy-related reasons within a year [7]. Patients with active cancer also exhibited significantly increased 6-month mortality compared to NCPs and patients with a remote history of cancer [18]. Additionally, any history of cancer was named the strongest independent predictor of 90-day mortality after MT, with ORs spanning from 7.8 in univariate and ~23 in multivariate analysis (NIHSS scores, hemorrhagic transformation, technical complications) of a mortality-predictor study [9].

Four studies reported no significant 90DM difference between CPs and NCPs [6,15,16,17], though one of them was missing data on the CPs [6]. One study was a 1-to-1 matched analysis of 24 AIS patients in each group, where the 90DM was more than double in the AM group, but the difference did not reach the significance threshold [17], and another study reported no difference in 90DM between patients with active cancer, remote cancer, and no cancer history; in both studies, cancer-related deaths were more frequent than stroke-related deaths in the AM groups [15,17].

One study also assessed the influence of metastatic disease on survival. No difference in 90DM between patients with or without metastatic disease was found exclusively in the MT group; for patients treated with IVT alone, 90DM was higher in patients with metastases [15].

### 3.3. Intracranial Hemorrhage

One of the most frequent complications of AIS and revascularization strategies is intracranial hemorrhage (ICH), asymptomatic or symptomatic (sICH), and hemorrhagic transformation (HT) of the infarct. As such, these occurrences were frequently examined in the included studies, as metrics of MT safety (Table 4, 2 CRS studies, see Section 3.6).

SICH was most commonly defined as a neurological worsening in terms of an increase in NIHSS (National Institutes of Health Stroke Scale) scores of ≥4, with radiological evidence of an ICH [17,23], though a definition was not always provided [16]. One study reported adherence to the Safe Implementation of Treatments in Stroke (SITS) criteria, defining sICH as a Type 2 parenchymal hemorrhage with clinical deterioration in NIHSS score of ≥4, or death [10], and six used the European–Australasian Cooperative Acute Stroke Study III Criteria, defining sICH as any extravasated blood in the parenchyma or within the cranium, that was the predominant cause of neurological deterioration in NIHSS score of ≥4, or death [8,11,15,17,18,19]. Most included studies reported no statistically significant differences in sICH between CPs and NCPs (one study assessed sICH only within 24 h [16]) [10,11,16,17,18,23], despite an occasional tendency towards increased rates in CPs [11]. One study reported higher sICH rates in AM patients, compared with patients with remote cancer or no cancer history, only in univariate analysis; in multivariate (adjusted for confounders such as age and baseline NIHSS), no significant differences were shown [15]. The presence of a metastatic disease did not increase sICH rates [15].

Regarding any kind of ICH, without differentiation between asymptomatic and symptomatic, or parenchymal and non-parenchymal, no significant differences between CPs and NCPs were reported in the majority of the relevant studies as well [8,10,11,17,20,22], with only one reporting significantly higher overall ICH rates (defined as subarachnoid or intraparenchymal), but no differences in regard to the type of ICH [13]. An additional CRS study also provided the rates of subarachnoid and parenchymal hemorrhages, showing no differences between CPs and NCPs [19], with these findings on the subarachnoid hemorrhages corroborated in a third study as well [8].

Hemorrhagic transformation, defined per the Heidelberg criteria [26], was reported as significantly higher in CPs in a 1:5 propensity-matched analysis [6]. HT was most frequently asymptomatic and did not significantly correlate to cancer stage or use of anticoagulants [6]. In the long-term outcome study of 12 CPs after MT, 3 patients exhibited HT (25%), and 1experienced a symptomatic subarachnoid hemorrhage [7]. One more study encompassing HT in their analysis of parenchymal hemorrhage also found no significant differences between patients with AM, remote cancer, and no cancer history [18].

The combination of IVT and MT did not seem to increase ICH risk either [10,15,18]. Craniectomy rates were also shown similar between CPs and NCPs [13,14].

### 3.4. Successful Reperfusion

The rates of successful reperfusion, most usually defined as a TICI (Treatment in Cerebral Infarction) score ≥2b [8,17,18,23], were also frequently assessed as a metric of MT efficacy.

Most studies reported no significant differences between CPs and NCPs [6,8,10,11,14,16,17,18,23], or when no comparison was made, successful reperfusion in the majority of the CPs (10/12 of CPs included) [7].

### 3.5. Functional Outcome

#### 3.5.1. NIHSS in the Acute Phase

The NIHSS is the scale most commonly used to assess IS patients in the acute phase, before and after a possible intervention, with higher scores (from 0 to 42) corresponding to more and greater neurological deficits. In the included studies, NIHSS scores at 24 [18,23] or 48 h [23] after admittance, at discharge [11], or baseline-to-discharge differences [6,10] were not significantly different between CPs and NCPs. Only one study reported significantly higher NIHSS scores in CPs [13].

#### 3.5.2. 90-Day mRS and NIHSS

As a metric of functional independence (FI), the modified Rankin Scale (mRS) was evaluated at 90 days (90DMRS) in several studies (Table 5, 3 CRS studies, see Section 3.6). MRS is a scale of 7 steps, accounting for the degree an individual can tend to their needs and everyday activities, with 0 corresponding to no symptoms, 5 corresponding to a bedridden patient that requires constant attention and care, and 6 meaning death. In most studies, an mRS score of 0–2 was usually named “functionally independent” or classified as a “good outcome” after stroke intervention [6,8,10,12,15,17,23].

CPs had worse 90DMRS scores [8,13,18,23] and were significantly less likely to be functionally independent than NCPs [13,16,23] in some studies, though this was not universally replicated, with more studies reporting no significant differences in 90DMRS [6] or functional independency, despite tendencies towards higher dependency rates in CPs [6,8,10,11,15,17,18]. The rates of CPs achieving functional independence in 3 months ranged from ~21% to ~47%, with more studies inclining towards values <30% [6,8,10,11,16,18,23]. Higher mRS scores were reported in patients with more advanced cancer disease, which is most likely due to the advanced stage of the malignancy itself [6]. MT was also associated with a shift towards worse 90DMRS in CPs [13].

CPs with a good pre-stroke condition (mRS of 0–1) also showed worse outcomes at 90 days compared to NCPs [23]. In a study of long-term out comes in 12 CPs, though an mRS score of 0–2 was achieved within 2 months for four patients, three of them deceased by the end of the year [7].

One study also compared NIHSS change at 3 months, reporting comparable changes in CPs and NCPs [6].

### 3.6. Cancer-Related Stroke

As previously mentioned, three studies assessed patients who demonstrated a cancer-related stroke, which seems to carry unique pathophysiological traits and stem from a state of generalized hypercoagulability [2].

In this setting, mortality at discharge and 90DM were both reported as significantly increased in the CRS group compared to large artery atherosclerosis (LAA) and cardioembolism (CE) [12], and non-CRS as a whole [14], while a technique-comparison study in 62 CRS patients reported a 90DM of almost 50% [19]. The causes of death at 3 months were not found significantly different between CRS and non-CRS, with most CRS deaths within 3 months being attributed to the AIS and occurring during hospitalization [14].

Regarding ICH, in one study, CRS patients exhibited significantly higher rates of any ICH [14], while in the other two, no comparisons with NCPs were made. In the first, only 1 in 19 CRS patients developed a sICH, while 4(21%) showed a HT [12], and in the second study, 15 in 62 (24.2%) CRS patients developed any hemorrhagic complication, with 8 of them being symptomatic [19]. Consequently, data on this patient subset pertaining to ICH remains inconclusive.

Moving on to reperfusion rates, only one study assessing CRS in particular reported that successful reperfusion was significantly less frequent in CRS compared to LAA and CE [12]; the second CRS study reported no difference between CRS and non-CRS strokes [14], while the third CRS study showed successful reperfusion for the majority of CRS patients [19].

Finally, CRS patients were also shown to have similar good outcome rates at discharge compared to LAA and CE, but significantly lower rates of a good outcome at 3 months compared to LAA and CE [12], NCPs [14], and overall (study with no comparison, only 17.7% of patients achieved functional independence) [19].

### 3.7. Other Parameters

Several studies included analyses on parameters that were not assessed in the other studies, but are of relevance regarding the prognosis of CPs with AIS undergoing MT, and will be presented in this section.

In the analysis of the MR CLEAN thrombectomy register, recurrent stroke occurred significantly more often in CPs, with an adjusted odds ratio of 3.1 [23]. A re-occlusion of the affected artery was also found in 3 out of 12 patients observed in the long-term outcome study of Oki et al. (2020) [7].

Expectedly, patients in a palliative setting showed higher mortality and lower functional independency rates in the study of Verschoof et al. (2022) [23]. This was reciprocated by Merlino et al. (2021), where patients with metastatic disease showed higher rates of functional dependency and sICH, although the values did not reach the significance threshold [15]. Interestingly, Pana et al. (2021) reported that patients with metastatic disease showed significantly higher chances of being discharged at home when receiving both IVT and MT, something not shown in patients without metastases [21].

Regarding patients with a remote history of cancer (not fulfilling the active cancer criteria), they did not seem to present any significant differences with patients with no cancer history at all, regarding complication rates and functional outcomes [11,15,18], though a tendency towards worse outcomes was noted in one study [15].

Rinaldo et al. (2019) also analyzed a small cohort of patients with brain malignancies, undergoing IVT or MT (10 in each modality) [22]. Both of these treatments were significantly less often performed in this subgroup, while, of those receiving MT, 1suffered an ICH, 4 died, and no one was discharged at home; interestingly, of the 10 receiving IVT, no one died and 2 were discharged at home. Mortality indices for this cancer subgroup were comparable with other malignancies.

Ozaki et al. (2021) reported similar reperfusion rates between CPs and NCPs, but when analyzing the different techniques separately, small-bore aspiration MT performed more poorly in comparison to stent retrievers [16]. On the contrary, Jeon et al. (2021) conducted an MT technique comparison study on CRS patients and reported that contact aspiration showed higher reperfusion rates and lower numbers of passes compared to stent retrievers [19].

## 4. Discussion

AIS is a common cerebrovascular event in cancer patients, and can even stem from cancer itself [2]. However, recanalization treatment modalities, such as IVT and MT, are less often administered to patients with concomitant malignancy, with IVT rates being more affected, due to numerous contraindications arising in this patient population [4]. The retrospective studies included in this review further corroborated the fact that IVT is administered to fewer CPs compared to NCPs, while MT seems to be performed in similar rates in both groups [16,22,23]. One study also highlighted the importance of certified stroke centers, showing that CPs received IVT at higher rates compared to non-certified centers, with comparable MT rates [22]. As such, further exploring the safety and efficacy of MT in CPs merits a more careful look, which was the aim of this review.

Concerning safety analyses, in-hospital mortality was not unanimously reported increased in cancer patients, with relatively equal numbers of studies, heterogeneity aside, finding it increased and not increased. The three studies that particularly examined CRS, showed significantly increased IHM and 90DM [12,14,19]. In this sense, the data regarding 90DM was more unanimous, with CPs treated with MT exhibiting greater long-term mortality rates in the majority of studies.

Before attributing the higher mortality to the intervention, the causes of death need to be more closely examined. In the acute phase and closely after the intervention, stroke-related deaths seem to be more prominent in CPs and CRS patients [6,7,14], a fact that possibly highlights that patients with malignancy are more susceptible to stroke and intervention complications, though interventional complications are comparable between CPs and NCPs [18]. However, malignancy-related deaths were increasingly reported in 90DM and 1-year analyses [7,12,14,15], highlighting how malignancy steadily becomes the major mortality driver after the acute phase. This could be due to several factors; patients usually need to stop their cancer treatment upon suffering an AIS, and the worsening of their general health status makes them susceptible to life-threatening conditions, such as infections and electrolyte disturbances. Nevertheless, one study did report, that though IHM was significantly increased in both IVT and MT groups, only for MT patients was the mortality index significantly increased [22]. On the other hand, one study reported that stroke-related deaths in CPs treated with MT with or without IVT were considerably fewer compared to CPs treated with IVT only [15]. As such, the data, especially regarding IHM, seem to be contradictory, and more studies with careful mortality analyses are needed.

Along this line of thought, the role of comorbidity factors must also be addressed. Cancer patients were unanimously reported to have lower rates of cerebrovascular risk factors than NCPs. Similarly, in the mortality predictor study by Awad et al. (2020) [9], no significant differences regarding hypertension, hyperlipidemia, atrial fibrillation, previous stroke history, as well as age were found between patients who survived MT and patients who did not; a history of cancer was, however, significantly less likely found in the survivors. As such, since other comorbid conditions do not seem to be more prevalent in CPs, the role of malignancy as the major mortality factor is further highlighted.

Furthermore, the rates of any or solely symptomatic ICH, and hemorrhagic transformation, did not show significantly higher rates in CPs for the majority of the studies. A 2021 meta-analysis also reported no association between active malignancy and sICH (5 studies, moderate heterogeneity) [27]. Even for CRS, the data available were not hinting towards an increased risk either, though the available studies were inconclusive. Similarly, successful reperfusion rates were similar between CPs and NCPs in all of the studies that included this metric, with only one study regarding CRS reporting lower rates compared to large-artery atherosclerosis and cardioembolism strokes [12]. As such, per the available data, it seems that MT in CPs is a safe modality.

However, the various MT techniques might also be of relevance. Ozaki et al. (2021) reported similar reperfusion rates between CPs and NCPs, but when analyzing the different techniques separately, the older generation, small-bore aspiration MT performed more poorly in comparison to stent retrievers [16]. This was not reported in the study of Jeon et al. (2021), where contact aspiration, whether alone or combined with a stent retriever, led to higher reperfusion rates and a smaller need for repeated passes [19]. This discrepancy may be attributed to technical differences in the used aspiration catheters, or center experience, and to the fact that Jeon et al. (2021) only examined patients with CRS. This correlates to thrombus composition; an interesting histopathology study on thrombi extracted from patients with active, inactive, and no cancer, showed that those extracted from CPs had significantly higher platelet and lower erythrocyte counts [28], and, as such, CRS thrombi may be susceptible to higher fragmentation and recurrence rates, and may thus respond differently to various techniques.

Regarding combination treatment (IVT and MT), it also seems to be safe in the setting of malignancy, or at least as safe as MT alone. The earliest study reported that there was no statistical difference in IHM between CPs and NCPs that received both [20]. This was reciprocated in the study of Pana et al. (2021), where neither non-metastatic nor metastatic cancer was associated with IHM in the MT group, regardless of whether the patients also received IVT or not, while on the contrary, IVT alone was associated with higher IHM in patients with metastatic disease; surprisingly, the treatment combination offered greater chances of at home discharge for metastatic patients [21]. Similarly, Merlino et al. (2021) found significantly increased 90DM rates for patients with active cancer treated with IVT, as opposed to those receiving MT with or without IVT [15]. Additionally, the combination of IVT and MT was not associated with higher hemorrhage rates in CPs in several studies as well [10,15,18].

As brain malignancy is considered a contraindication for thrombolysis [5], and MT is also more reluctantly performed in this CP subgroup, data regarding stroke treatment in brain malignancy are even scarcer. Ozaki et al. (2019) analyzed a small subgroup of patients with brain malignancies that received either IVT or MT and reported comparable mortality rates with other malignancies, but worse mortality and at-home-discharge rates for MT compared to IVT [22]. There are limited data regarding the safety and efficacy of reperfusion strategies in this population, especially regarding MT, so these patients represent important dilemmas when otherwise suitable for acute intervention, and more research and information on this subject is urgently needed.

Of note, the various cancer types were also not separately analyzed in most of the included studies, most likely due to the relatively limited sample sizes. Solely Joshi et al. (2021) performed a comparison between patients with hemorrhagic transformation and not, and reported no differences in the various cancer types and stages [6]. Regarding metastatic disease, the available data are also scarce, and expectedly hints towards an influence of metastatic disease on long-term survival, though no statistically significant differences have been shown [15,21].

On to the second objective, the efficacy of MT in CPs, despite most studies not finding statistically significant differences in functional independence three months after the intervention, there was a unanimous trend towards worse outcomes, and the CPs that achieved functional independence were usually less than one-third of the whole. This percentage is comparable to NCPs that did not receive MT in the MR CLEAN trial [29], or to patients beyond the age of 80 [30]. One could argue that this percentage does not suffice to warrant a high-cost and potentially dangerous procedure, such as MT. However, the data available show that it is safe in this population, and denying the chance of adequate recovery in CPs otherwise suited for the procedure could be considered ‘unethical’, especially knowing that AIS in this population is in itself associated with more complications and worse outcomes, such as prolonged hospitalizations and fewer routine discharge rates regardless of metastatic disease [21]. Additionally, some studies reported independence rates in almost half of the treated patients [6], providing even greater ‘encouragement’ towards MT.

The available literature may not suffice to undoubtedly prove the efficacy of MT in CPs yet, nevertheless, it was a common author conclusion that despite the potential long-term unfavorable prognosis, MT could be considered in the hopes of offering a chance for functional recovery. Of note, post-procedural ICH and higher NIHSS scores have been repeatedly associated with worse functional outcomes and overall prognosis [8,10,13,14,31], though the evidence for NIHSS was stronger (independent predictors of outcome in adjusted analysis [14] and SICH [15], or maintained significance [10]). It could thus be claimed that the higher functional dependency rates noted in some studies are associated with the heavier strokes that CPs experience, and not with treatment per se.

Finally, CRS, most commonly stemming from cancer-associated hypercoagulability, is often classified as a stroke of “cryptogenic origin”, and seems to be associated with a worse prognosis, as reflected by the heightened mortality, possible higher ICH rates, and worse functional outcomes, despite timely and successful intervention [12,14,19]; in fact, the lowest 90D functional independence percentage (17.7%) was reported in a CRS study [19]. The unfavorable outcome in this subgroup is further evinced by studies that showed better outcomes and reduced mortality in CPs with strokes of determined etiology [18], or by the sole 1-year survivor of the long-term outcome study of Oki et al. (2020), whose AIS etiology was infectious endocarditis [7], and not non-bacterial thrombotic endocarditis, a common and often detrimental manifestation of cancer-associated hypercoagulability [2]. These unfavorable metrics might again raise the question of MT’s utility in CRS patients, and it is important to consider that these patients are particularly susceptible to complications and stroke-related deaths in the acute phase [14], while they also deteriorate quickly and die mostly due to malignancy-related reasons within 3 months [12].However, MT might be their only therapeutic choice, given that IVT contraindications are particularly frequent in this patient sample, and may indeed improve the quality of their remaining lifespan. Nevertheless, the available evidence remains scarce and the overall condition of the patient should be taken into consideration, so that the proposed intervention and possible outcomes align with the patient’s goal of care.

In this light, secondary or even primary prevention should also be brought to attention. Malignancy is a well-known predisposing factor for AIS [2] and might require more aggressive preventive strategies than previously thought. For instance, Oki et al. (2020) reported that half of their AIS patients were on anticoagulants (direct oral anticoagulants or warfarin) when the AIS occurred, and one-third experienced a re-occlusion of the affected artery shortly after the intervention [7]. Regarding the use of anticoagulants at least, the literature and guidelines so far seem to agree that direct oral anticoagulants are safer and more effective than warfarin in cancer patients, so this anticoagulant class, when not contraindicated, should be chosen [4]. However, the necessity and safety of anticoagulation in cancer-related coagulopathy are still under discussion and anticoagulation administration is usually based on the respective guidelines of each entity the coagulopathy causes, for example deep vein thrombosis or pulmonary embolism. For regular IS prevention, platelet aggregation inhibitors (aspirin, clopidogrel, etc.) do not present any particular indications or contraindications for cancer patients and should be considered [5,32].

## 5. Conclusions

Conclusively, per the limited available data, in-hospital mortality and peri-interventional complications do not seem to be increased in cancer patients receiving MT; the increased mortality after the acute phase seems to be more heavily influenced by the malignancy itself, and not the intervention, though evidence is still lacking. As such, MT, also in combination with IVT, seems to be a safe treatment modality in this population. Regardless of the tendency for a worse prognosis, considerable proportions of CPs regained functional independence in 90 days [23], and given that CPs tend to have worse outcomes after IS regardless of treatment or not [22], with some studies even claiming that thrombolytic treatments offset this worse prognosis [21], we conclude that revascularization therapies and MT, in particular, can be offered to AIS patients with concomitant or even underlying malignancy. However, in light of the limited evidence and lack of high-quality prospective studies, it is still uncertain whether MT can be applied to patients with all kinds of malignancies and malignancy stages, given the worse overall outcomes of patients with advanced disease, so an individualized approach should be undertaken for every patient. More research towards this direction is more than warranted.

## Figures and Tables

**Figure 1 jcm-11-04696-f001:**
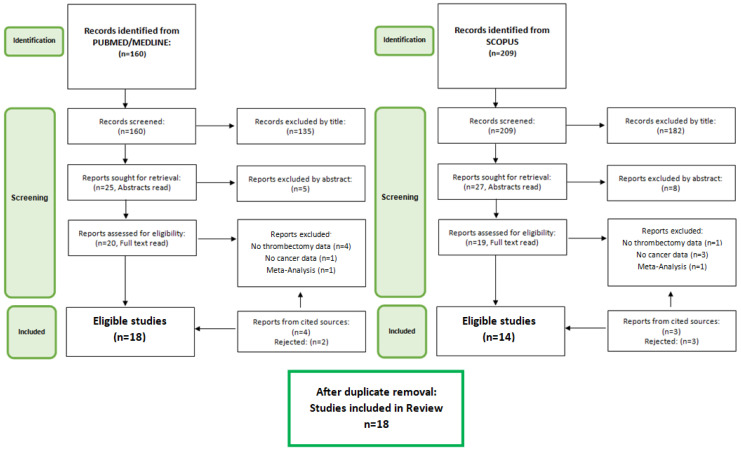
Literature search flowchart.

**Table 1 jcm-11-04696-t001:** Included Studies, Study Characteristics, and Cancer Patient numbers.

Author, Year	Patient Numbers	Study Characteristics
Awad et al., 2020 [9]	14 CP/111 MTs * (12.6% **)	MT files, mortality predictors study, any history of cancer
Cho et al., 2020 [10]	27 CP/378 MTs (7.2%)	MT files, active CP vs. NCP *^3^
Ciolli et al., 2021 [11]	14 CP/305 MTs (4.5%)	MT files, active CP vs. NCP (+/− history of cancer)
Jeon et al., 2021 [19]	62 CRS	MT files, MT technique comparison in CRS
Joshi et al., 2022 [6]	19 CP/95 matched NCP	MT files, 1:5 propensity-matched analysis
Jung et al., 2018 [12]	19 CRS/329 MTs (5.7%)	MT files, CRS vs. non-CRS
Lee et al., 2019 [13]	26 CP/253 MTs (10.2%)	MT files, active CP vs. NCP, history of cancer excluded
Lee et al., 2021 [14]	34 CRS/341 MTs (9.9%)	MT files, CRS vs. non-CRS
Mattingly et al., 2022 [8]	25 CP/284 MTs (8.8%)	MT files, active CP vs. NCP
Merlino et al., 2021 [15]	21 CP *^4^/173 MTs (12.1%)	MT files, active CP vs. NCP vs. remote CP
Murthy et al., 2013 [20]	193CP/6766 IVT+MTs (2.8%)	IVT files, ICD codes, IVT+MT, active CP vs. NCP
Oki et al., 2020 [7]	12 CP/124 MTs (9.6%)	MT files, active CP vs. NCP
Ozaki et al., 2021 [16]	19 CP/300 MTs (6.3%)	MT files, active CP vs. NCP
Pana et al., 2021 [21]	1330 CP/34,420 MTs (3.8%)	IS files, ICD codes, active CP +/− metastasis vs. NCP
Rinaldo et al., 2019 [22]	857 CP/17,268 MTs (4.9%)	IS files, ICD codes, active CP vs. NCP
Sallustio et al., 2019 [17]	24 CP/24 matched NCP	MT files, 1:1 matched analysis
Verschoof et al., 2022 [23]	124 CP/2583 MTs (4.8%)	MR CLEAN Registry, active CP vs. NCP
Yoo et al., 2021 [18]	42 CP *^4^/685 MTs (6.1%)	MT files, active CP vs. NCP vs. remote CP

* MTs: all MT patients. ** Data was available for 111 out of 134 MTs, a percentage possibly not representative of the CP fraction. *^3^ Non-cancer patients. *^4^ Only patients with active cancer.

**Table 2 jcm-11-04696-t002:** Overview of baseline characteristics and cerebrovascular risk factors in cancer patients compared to non-cancer patients.

Author, Year	Age	Sex (M/F)	AF	DM	HT	History of AIS/TIA	HL	Smoking	CAD
Cho et al., 2019 [10]	-	M↑	-	-	-	-	-	-	NA
Ciolli et al., 2021 [11]	-	-	NA	-	-	NA	↓	NA	NA
Jung et al., 2018 [12]	-	M↓	↓	-	↓	-	-	↓	-
Lee et al., 2019 [13]	↓	-	↓	-	-	-	-	NA	NA
Lee et al., 2021 [14]	-	-	↓	-	-	-	-	-	-
Mattingly et al., 2022 [8]	-	-	NA	NA	NA	NA	NA	NA	NA
Merlino et al., 2021 [15]	-	-	-	-	-	-	-	-	-
Ozaki et al., 2021 [16]	-	M↓	-	-	-	NA	NA	NA	NA
Pana et al., 2021 [21]	-	M↓	NA	↓	-	-	-	-	-
Sallustio et al., 2019 [17]	-	-	-	-	-	NA	NA	-	NA
Verschoof et al., 2022 [23]	-	-	-	-	↓	-	-	↑	-
Yoo et al., 2021 [18]	↓	-	↓	-	-	NA	-	-	-

Note: -: No significant difference. ↑: Increased in cancer patients. ↓: Decreased in cancer patients.NA: Not assessed/applicable. M/F: Male/Female, M: Male. AF: Atrial fibrillation. DM: Diabetes mellitus. HT: Hypertension. TIA: Transient ischemic attack. HL: Hyperlipidemia. CAD: Coronary artery disease.

**Table 3 jcm-11-04696-t003:** Overview of in-hospital and 90-day mortality tendency in cancer patients compared to non-cancer patients.

Author, Year	In-Hospital Mortality	90-Day Mortality
Awad et al., 2020 [9]	NA	↑
Cho et al., 2020 [10]	-	↑
Ciolli et al., 2021 [11]	↑	↑
Jeon et al., 2021 [19]	NA	↑ *
Joshi et al., 2022 [6]	NA	↑ **
Jung et al., 2018 [12]	↑	↑
Lee et al., 2019 [13]	NA	↑
Lee et al., 2021 [14]	↑	↑
Mattingly et al., 2022 [8]	NA	↑
Merlino et al., 2021 [15]	-	-
Murthy et al., 2013 [20]	-	NA
Oki et al., 2020 [7]	↑ *	↑ *
Ozaki et al., 2021 [16]	NA	-
Pana et al., 2021 [21]	-	NA
Rinaldo et al., 2019 [22]	↑	NA
Sallustio et al., 2019 [17]	-	↑ *^3^
Verschoof et al., 2022 [23]	↑	↑

Note: -: No significant difference. ↑: Increased in cancer patients. NA: Not assessed/applicable. * Study did not compare CP and NCP, increased based on respective NCP metrics of other similar studies. ** Not statistically significant, missing patient data. *^3^ 1:1 matched analysis of 24 CPs, more than double 90DM but not statistically significant.

**Table 4 jcm-11-04696-t004:** Overview of symptomatic and any ICH, and hemorrhagic transformation tendencies in cancer patients compared to non-cancer patients.

Author, Year	sICH	Any ICH	HT
Cho et al., 2020 [10]	-	-	NA
Ciolli et al., 2021 [11]	-	-	NA
Jeon et al., 2021 [19]	↑ *	↑ *	NA
Joshi et al., 2022 [6]	NA	NA	↑
Lee et al., 2019 [13]	NA	↑	NA
Lee et al., 2021 [14]	NA	↑	-
Mattingly et al., 2022 [8]	NA	-	NA
Merlino et al., 2021 [15]	↑/-	NA	NA
Murthy et al., 2013 [20]	NA	-	NA
Oki et al., 2020 [7]	NA	NA	↑ *
Ozaki et al., 2021 [16]	-	NA	NA
Rinaldo et al., 2019 [22]	NA	-	NA
Sallustio et al., 2019 [17]	-	-	NA
Verschoof et al., 2022 [23]	-	NA	NA
Yoo et al., 2021 [18]	-	NA	-

Note: -: No significant difference. ↑: Increased in cancer patients. NA: Not assessed/applicable.* Study did not compare CP and NCP, increased based on respective NCP metrics of other similar studies.

**Table 5 jcm-11-04696-t005:** Overview of 90-daymRS scores and functional independence tendencies in cancer patients compared to non-cancer patients.

Author, Year	90D MRS	90D FI
Cho et al., 2020 [10]	NA	-
Ciolli et al., 2021 [11]	NA	-
Jeon et al., 2021 [19]	↓ *	↓ *
Joshi et al., 2021 [6]	-	-
Jung et al., 2018 [12]	NA	↓
Mattingly et al., 2022 [8]	↓	-
Merlino et al., 2021 [15] **	NA	-
Lee et al., 2019 [13]	↓	↓
Lee et al., 2021 [14]	↓	↓
Ozaki et al., 2021 [16]	-	↓
Sallustio et al., 2019 [17]	NA	-
Verschoof et al., 2022 [23]	↓	↓
Yoo et al., 2021 [18]	↓	-

Note: -: No significant difference.↓: Decreased in cancer patients. NA: Not assessed/applicable. * Study did not compare CP and NCP, increased based on respective NCP metrics of other similar studies. ** Study compared functional dependency, defined as mRS 3–5, FI rates deducted from the available data.

## Data Availability

Not applicable.

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
