# Peer review of "Mechanical Thrombectomy for Acute Ischemic Stroke in Patients with Malignancy: A Systematic Review"

_jcm, 2022, doi:10.3390/jcm11164696_

Round 1
Reviewer 1 Report
It is my great pleasure to review this manuscript. The authors aimed to review and summarize current evidence on the safety and efficacy of mechanical thrombectomy (MT) in patients with cancer. It is admirable for the authors to look into the current literature and advocate for potential treatment options and recovery opportunities for these patients. However, the conclusion “MT must be considered” or “should be offered” is not well-supported by the current evidence. As the authors mentioned, all studies were retrospective and contained significant heterogeneity (e.g., active vs. past cancer, primary vs. metastatic cancer, whether the stroke was considered related to cancer; additionally, cancer type and stages were not stratified). Although MT appears to be safe in certain cancer patients, it is truly unclear whether it is applicable to all types of cancer patients and whether it actually brings short-term/long-term benefits to all. Without well-designed prospective studies, it is very difficult to control confounders and biases to support such strong statements. A universal strong suggestion to provide MT may not be appropriate at this time. Acute ischemic stroke, especially when concurrent with aggressive cancer, warrants individualized assessment (e.g., based on stroke size, severity, cancer type and stage, premorbid functions) and careful discussions on treatment options to align patients’ goals of care. Would suggest rephrasing relevant statements. The manuscript is overall well-written except for minor typos and formatting errors.
For the tables:
1. Please add an annotation for “M↑” or “M↓” in Table 2.
2. What is the difference between the mark “-” and “NA” in the tables? Does “-” also mean “not assessed/applicable”?
Thanks!
Author Response
We cordially thank the Reviewers for taking the time to assess our manuscript and provide valuable criticism. We have taken all of the comments into careful consideration and adjusted the manuscript accordingly. Below, we have replied to the comments individually. We hope that the manuscript is now deemed fit for publication.
REVIEWER 1
It is my great pleasure to review this manuscript. The authors aimed to review and summarize current evidence on the safety and efficacy of mechanical thrombectomy (MT) in patients with cancer. It is admirable for the authors to look into the current literature and advocate for potential treatment options and recovery opportunities for these patients. However, the conclusion “MT must be considered” or “should be offered” is not well-supported by the current evidence. As the authors mentioned, all studies were retrospective and contained significant heterogeneity (e.g., active vs. past cancer, primary vs. metastatic cancer, whether the stroke was considered related to cancer; additionally, cancer type and stages were not stratified). Although MT appears to be safe in certain cancer patients, it is truly unclear whether it is applicable to all types of cancer patients and whether it actually brings short-term/long-term benefits to all. Without well-designed prospective studies, it is very difficult to control confounders and biases to support such strong statements. A universal strong suggestion to provide MT may not be appropriate at this time. Acute ischemic stroke, especially when concurrent with aggressive cancer, warrants individualized assessment (e.g., based on stroke size, severity, cancer type and stage, premorbid functions) and careful discussions on treatment options to align patients’ goals of care. Would suggest rephrasing relevant statements.
Reply: We thank the Reviewer for this comment and agree that MT at this point does not carry evidence strong enough in this patient subgroup. As such, we have adjusted all the respective sections, especially the discussion, to not include this absolute statement and highlight the need for more research and individualized approaches.
The manuscript is overall well-written except for minor typos and formatting errors.
Reply: The manuscript was thoroughly re-read and any errors were corrected.
For the tables:
- Please add an annotation for “M↑” or “M↓” in Table 2.
- What is the difference between the mark “-” and “NA” in the tables? Does “-” also mean “not assessed/applicable”?
Reply: The “-“ symbol denotes no differences between patients with and without cancer. In order to avoid confusion, we have added annotations under every table with symbols, in a form of an appendix to facilitate readability. We thank the Reviewer for the remark.

Reviewer 2 Report
The authors reviewed 18 retrospective studies regarding MT in cancer patients (CPs). They found rates of in-hospital mortality, intracranial hemorrhage of any kind, reperfusion rates, and discharge condition did not seem to present any considerable differences between CPs and patients without cancer while 90-day mortality was higher and 90-day functional independence was lower in CPs. Therefore, they concluded that MT appears to be a safe treatment option for CPs.
This review evaluated the safety and efficacy of MT in cancer patients based on recent studies and proposed the suggestion of MT treatment in CPs. A few comments are listed as following:
1. The results and discussion about "cancer-related stroke" (CRS) could be divided by its own section since there are already clear relationship between malignancy and Acute Ischemic Stroke. In addition, the studies involving patients with CRS exhibited obviously increased In-Hospital Mortality and the rates of Any ICH, as well as reduced 90D FI, indicating systematic differences between patients with CRS and other CPs.
2. More evidence are needed to conscience the conclusion “The 90-day mortality and outcome rates are commonly attributed to the malignancy and not to the intervention”. Do any reviewed studies evaluate the long-term recovery from MT in CPs and NCPs, and whether there is a linkage between that and reduced 90D FI in CPs?
Author Response
The authors reviewed 18 retrospective studies regarding MT in cancer patients (CPs). They found rates of in-hospital mortality, intracranial hemorrhage of any kind, reperfusion rates, and discharge condition did not seem to present any considerable differences between CPs and patients without cancer while 90-day mortality was higher and 90-day functional independence was lower in CPs. Therefore, they concluded that MT appears to be a safe treatment option for CPs.
This review evaluated the safety and efficacy of MT in cancer patients based on recent studies and proposed the suggestion of MT treatment in CPs. A few comments are listed as following:
- The results and discussion about "cancer-related stroke" (CRS) could be divided by its own section since there are already clear relationship between malignancy and Acute Ischemic Stroke. In addition, the studies involving patients with CRS exhibited obviously increased In-Hospital Mortality and the rates of Any ICH, as well as reduced 90D FI, indicating systematic differences between patients with CRS and other CPs.
Reply: We thank the Reviewer for this insight. There was a significant distinction in this subset of patients, so a different subsection is reasonable. As such, we have proceeded to move the results of the studies assessing these patients in a standalone section, and have enriched the discussion CRS section. In order to maintain the integrity of the tables presenting the entirety of the relevant studies, we left the tables in their current state, and clarified that a number of those shown pertain to CRS and the results are presented in the respective section.
- More evidence are needed to conscience the conclusion “The 90-day mortality and outcome rates are commonly attributed to the malignancy and not to the intervention”. Do any reviewed studies evaluate the long-term recovery from MT in CPs and NCPs, and whether there is a linkage between that and reduced 90D FI in CPs?
Reply: All the relevant results from studies are reported in this review; more long-term assessments have not been made available. The statement was made due to statements from the studies’ authors. However, we do agree that more evidence is needed and have proceeded to adjust it accordingly. It now reads as “Per the available evidence, MT appears to be a safe treatment option for CPs. It is still unclear whether the 90-day mortality and outcome rates are more heavily influenced by the malignancy and not the intervention itself, so MT can be considered in CPs with prospects of a good functional recovery, undertaking an individualized approach.”
